# Learning incomplete factorization preconditioners for GMRES

Paul Häusner[*1], Aleix Nieto Juscafresa[†1], and Jens Sjölund[1]

[1]Department of Information Technology, Uppsala University, Sweden
{paul.hausner, aleix.nieto-juscafresa, jens.sjolund}@it.uu.se

## Abstract

Incomplete LU factorizations of sparse matrices are widely used as preconditioners in Krylov subspace methods to speed up solving linear systems. Unfortunately, computing the preconditioner itself can be time-consuming and sensitive to hyper-parameters. Instead, we replace the hand-engineered algorithm with a graph neural network that is trained to approximate the matrix factorization directly. To apply the output of the neural network as a preconditioner, we propose an output activation function that guarantees that the predicted factorization is invertible. Further, applying a graph neural network architecture allows us to ensure that the output itself is sparse which is desirable from a computational standpoint. We theoretically analyze and empirically evaluate different loss functions to train the learned preconditioners and show their effectiveness in decreasing the number of GMRES iterations and improving the spectral properties on synthetic data. The code is available at https://github.com/paulhausner/neural-incomplete-factorization.

## 1 Introduction

The generalized minimal residual method (GMRES) [1] is one of the most popular iterative methods to solve large-scale and sparse linear equation systems of the form $\boldsymbol{Ax} = \boldsymbol{b}$. Throughout the paper, we assume the square system matrix $\boldsymbol{A}$ to be real-valued and full rank. Therefore, the unique solution to the equation system is given by $\boldsymbol{A}^{-1}\boldsymbol{b}$. However, inverting the matrix directly, or equivalently computing a full matrix factorization, scales computationally poorly and suffers from numerical instabilities. Instead, iterative Krylov subspace methods, such as GMRES, which refine an approximation of the solution in each step are the most common solving technique for large-scale and sparse matrices. The convergence of these methods depends on the conditioning and singular value distribution of the system matrix and the right-hand side $\boldsymbol{b}$. Therefore, the choice of appropriate preconditioner – designed to improve the system properties – is critical to obtain a fast and accurate solution for the equation system [2].

One of the most common choices of preconditioners is the incomplete LU factorization (ILU). As the name suggests, the matrix $\boldsymbol{A}$ is factorized into lower ($\boldsymbol{L}$) and upper ($\boldsymbol{U}$) triangular factors allowing an efficient matrix inversion. To decrease the storage and computational time not all elements of the factorization are computed for the preconditioner leading to a sparse but incomplete factorization. The computation of the preconditioner is time-consuming, difficult to parallelize, and can suffer from numerical instabilities [3, 4]. In this work, we replace the numerical computation of the incomplete LU factors with a graph neural network (GNN). The GNN learns to predict the corresponding incomplete factors of the matrix directly by training against data. The main contributions of our paper are the following:

- We design a GNN architecture that outputs a sparse and non-singular incomplete LU factorization.

- We theoretically analyze existing loss functions from the literature and derive their connection to large and small singular values.

- Building on these insights, we derive a novel loss function for training learned preconditioners that combines the benefits of prior approaches.

In numerical experiments, we validate the effectiveness of our model in reducing GMRES iterations in combination with the different loss functions and validate the theoretical results on a synthetic dataset.

**Related work** Most similar to our work, Häusner et al. [5] and Li et al. [6] both learn an incomplete factorization for the conjugate gradient method. Trifonov et al. [7] extend this to combine data-driven and classical algorithms by correcting the output of the incomplete Cholesky factorization with a learned component. However, all of these methods assume the matrix to be symmetric and positive definite and utilize the incomplete Cholesky factorization as the underlying preconditioning technique. In contrast, we rather focus on the more general incomplete LU factorization, which does not require positive definiteness or symmetry, within the GMRES algorithm similar to the recently proposed GraphPAN [8].

In previous work, Chen [9] instead proposes directly estimating the inverse of the matrix using a non-linear neural network as a preconditioner, requiring the use of the flexible GMRES method, which

---

[*]Joint first and corresponding author.
[†]Joint first author.

Proceedings of the 6th Northern Lights Deep Learning Conference (NLDL), PMLR 265, 2025.

has a more complex convergence behavior. However, the proposed method requires retraining for each new problem. In contrast, we propose leveraging the similarity within a class of linear equation systems to first train the model offline and then generate preconditioners for new problems with a negligible computational overhead during inference time.

For the class of sparse approximate inverse preconditioners, Bånkestad et al. [10] learn the sparsity pattern of the approximate inverse matrix for which the preconditioner can be computed efficiently. Li et al. [11] instead propose to learn auto-encoder-based generative models to generate preconditioners. In this paper, we focus instead on learning factorized preconditioners that allow easy inversion rather than learning the inverse matrix directly.

## 2  Background

We start by providing a brief overview of the GM-RES algorithm, emphasizing the critical role of preconditioners in accelerating convergence. Then, we describe graph neural networks which parameterize the mapping to learn data-driven preconditioners.

### 2.1  GMRES algorithm

We focus on GMRES [1], a popular and widely adopted iterative Krylov subspace method to solve general linear equation systems of the form $\boldsymbol{Ax} = \boldsymbol{b}$ where $\boldsymbol{A}$ is a $n \times n$ non-singular and real valued matrix, i.e. $\boldsymbol{A} \in \mathrm{GL}(n, \mathbb{R})$. The core idea of Krylov subspace methods is to iteratively refine an approximation of the solution to the problem starting from an initial guess $\boldsymbol{x}_0$. At each iteration, the approximate solution is computed by minimizing the Euclidean norm of the residual vector within the corresponding Krylov subspace:

$$\boldsymbol{x}_k = \operatorname*{arg\,min}_{\boldsymbol{x} \in \boldsymbol{x}_0 + \mathcal{K}_k(\boldsymbol{A}, \boldsymbol{r}_0)} \|\boldsymbol{b} - \boldsymbol{Ax}\|_2, \qquad (1)$$

where the $k$-th Krylov subspace $\mathcal{K}_k(\boldsymbol{A}, \boldsymbol{r}_0) = \operatorname{span}\{\boldsymbol{r}_0, \boldsymbol{Ar}_0, \boldsymbol{A}^2\boldsymbol{r}_0, \dots, \boldsymbol{A}^{k-1}\boldsymbol{r}_0\}$ of $\boldsymbol{A}$ is generated by the initial residual $\boldsymbol{r}_0 = \boldsymbol{b} - \boldsymbol{Ax}_0$ [2]. By adding elements $\boldsymbol{A}^i\boldsymbol{r}_0$ to the subspace basis, the dimensionality of the Krylov subspace increases with each iteration.[1] This allows GMRES to further reduce the minimal residual norm in Equation (1) with more iterations. The GMRES algorithm is guaranteed to find the exact solution to the original linear equation system in at most $n$ steps.

Given the previous solution $\boldsymbol{x}_k$ the next iterate can be computed efficiently by solving a reduced unconstrained system in the subspace. Each iteration consists of two main steps. In the first step of the

algorithm, an orthonormal basis for the subspace $\mathcal{K}_k$ is computed. This can be achieved, for example, using Arnoldi's algorithm which is based on the Gram-Schmidt procedure. In the second step, an unconstrained least squares problem is solved via the QR factorization of the subspace basis obtained in the first step. Based on these two steps, the solution of Equation (1) can be constructed efficiently. The detailed algorithm is shown in Appendix A.

**Preconditioning** Arguably, the most important design choice for any Krylov subspace method is the choice of preconditioner [3]. The goal of preconditioning is to replace the original linear equation system with a new preconditioned system that exhibits a better clustering of singular values, which in turn can lead to faster convergence of the iterative scheme [12]. For a non-singular and easy-to-invert matrix $\boldsymbol{P}$, the linear equation system $\boldsymbol{AP}^{-1}\boldsymbol{y} = \boldsymbol{b}$ is solved instead of the original problem. The solution to the original problem is then given by $\boldsymbol{x} = \boldsymbol{P}^{-1}\boldsymbol{y}$.[2]

Constructing a preconditioner involves trading off the time required to compute the preconditioner itself and the resulting speedup in the iterative scheme [3]. Since the original system $\boldsymbol{A}$ is typically sparse, we restrict the preconditioning matrix $\boldsymbol{P}$ to be subject to sparsity constraints as well.

Algebraic preconditioners do not assume any additional problem information and compute the preconditioning matrix solely based on the structure and values of the matrix $\boldsymbol{A}$. In contrast, geometric methods take into account the underlying problem domain. In the simplest case, the Jacobi preconditioner approximates the matrix $\boldsymbol{A}$ using a diagonal matrix. More advanced methods such as incomplete factorizations or the Gauss-Seidel method compute approximate factorizations of the original system $\boldsymbol{A}$ which implicitly give rise to preconditioners as they allow efficient inversions [3, 13]. Most commonly, the approximation utilizes triangular factorizations since efficient inversion in at most $\mathcal{O}(n^2)$ operations can be achieved and sparsity can be exploited [14]. However, It is also possible to approximate the preconditioning matrix $\boldsymbol{P}^{-1}$ directly [15].

### 2.2  Graph neural networks

Graph neural networks (GNNs) are a recently popularized family of neural network architectures designed to process data represented on unstructured grids or graphs [16].

A graph is a tuple $\mathcal{G} = (\mathcal{V}, \mathcal{E})$ consisting of vertices (or nodes) $\mathcal{V}$ and a bivariate relation of edges $\mathcal{E} \subseteq \mathcal{V} \times \mathcal{V}$. With slight abuse of notation, we refer to the edge features of the edge $e = (i, j) \in \mathcal{E}$ as $\boldsymbol{e}_{ij} \in \mathbb{R}^m$ and node features of node $i \in \mathcal{V}$ as

---

[1]If the vector is linearly dependent to the basis the dimensionality remains the same but the exact solution to the linear equation system can be found within the existing subspace.

[2]Here, we use right preconditioning, other formulations use left- or split-preconditioning instead and often lead to similar but not equivalent results.

$\boldsymbol{n}_i \in \mathbb{R}^k$. The message-passing framework updates the initial edge and node features – representing the input data – in each layer $l$ as follows:

$$\boldsymbol{e}_{ij}^{l+1} = \psi_{\boldsymbol{\theta}}(\boldsymbol{e}_{ij}^l, \boldsymbol{n}_i^l, \boldsymbol{n}_j^l), \tag{2a}$$

$$\boldsymbol{m}_i^{l+1} = \bigoplus_{j \in \mathcal{N}(i)} \boldsymbol{e}_{ji}^{l+1}, \tag{2b}$$

$$\boldsymbol{n}_i^{l+1} = \phi_{\boldsymbol{\theta}}(\boldsymbol{n}_i^l, \boldsymbol{m}_i^{l+1}). \tag{2c}$$

Here, the functions $\psi$ and $\phi$, which update the node and edge features respectively, are parameterized by neural networks, and their respective parameters $\boldsymbol{\theta}$ are learned during the model training. Note that all feature vectors within a single layer must be of the same size, although the embedding size can vary between different layers. Equation (2b) shows the aggregation of the adjacent edges features for each vertex $i \in \mathcal{V}$ called the neighborhood $\mathcal{N}(i)$. Any permutation invariant function $\oplus$, such as sum and mean, can be utilized for this aggregation step [17].

# 3 Method

Throughout this paper, we are interested in solving linear equation systems that share an underlying structure. Instead of having direct access to a probability distribution over the problems from a specific problem class, we obtain a dataset $\mathcal{D}$ with a finite number of i.i.d. samples $\boldsymbol{A}_1, \boldsymbol{A}_2, \ldots \boldsymbol{A}_{|\mathcal{D}|}$. As previously stated, we assume each $\boldsymbol{A}_i$ to be square, non-singular and sparse.

## 3.1 Network architecture

The goal of this paper is to learn a mapping $f_{\boldsymbol{\theta}}$, parameterized by a graph neural network, that takes a non-singular matrix $\boldsymbol{A} \in \mathrm{GL}(n, \mathbb{R})$ as input and outputs the corresponding preconditioner $\boldsymbol{P}$ for the equation system. In order to ensure that the preconditioning system is easily invertible – such that we can apply it within the GMRES algorithm – we learn the LU factorization of the preconditioner $\boldsymbol{P}$ instead, by mapping $\boldsymbol{A}$ to two real matrices $\boldsymbol{L}$ and $\boldsymbol{U}$. The output matrices are lower and upper triangular, respectively, with non-zero diagonal entries to guarantee their invertibility by utilizing a suitable activation function. For notational simplicity, we do not explicitly mention the dependence of the factors on the original matrix $\boldsymbol{A}$ and the neural network parameters $\boldsymbol{\theta}$ when the relationship is clear from the context.

**Sparsity** The two output factors, $\boldsymbol{L}$ and $\boldsymbol{U}$, are subject to sparsity constraints. Similar to incomplete LU factorization methods without fill-ins, ILU(0), we enforce the same sparsity patterns to the elements of $\boldsymbol{L}$ and $\boldsymbol{U}$ as the original matrix $\boldsymbol{A}$. This sparsity constraint is directly encoded in the graph neural network architecture.

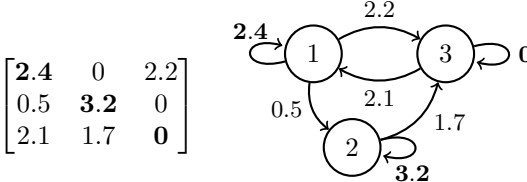

**Figure 1.** Non-symmetric matrix $\boldsymbol{A}$ (left) and the corresponding sparse Coates graph representation (right). Additionally to the non-zero elements in the matrix, the graph has been modified by adding edges for all missing diagonal elements, corresponding to self-loops.

**Network parameterization** To parameterize the function $f_{\boldsymbol{\theta}}$ that outputs the preconditioner, we exploit the strong connection of matrices and linear algebra with graph neural networks [18, 19].

We treat the system $\boldsymbol{A}$ as the adjacency matrix of a corresponding graph that we use directly within the message-passing framework. This transformation is known in the literature as Coates graph representation is depicted in Figure 1 [20]. Similar to Häusner et al. [5] and Tang et al. [21], we use node features describing the structural properties of the corresponding rows and columns of the matrix. We use the non-zero elements of $\boldsymbol{A}$ as the input edge features of the graph.

Compared to previous approaches which only output a single lower triangular factor, our model outputs two separate matrices $\boldsymbol{L}$ and $\boldsymbol{U}$. Häusner et al. [5] incorporate positional encoding by modifying the Coates graph representation, allowing them to output the lower triangular factor as the message passing is only executed over this subset of edges. In contrast, we add positional encoding directly to the input data by adding an edge feature to each directed edge in the graph. This feature indicates whether the final output embedding of the corresponding edge belongs to the upper- or lower-triangular part ($\pm 1$ respectively) without changing the underlying graph structure used for message passing.

**Invertibility** To obtain a valid preconditioner for the GMRES method during inference, the predicted factors $\boldsymbol{L}$ and $\boldsymbol{U}$ from the model's forward pass must be invertible. Therefore, we need to enforce non-zero elements on each diagonal.

Since the input matrix $\boldsymbol{A}$ is not guaranteed to have non-zero diagonal elements, we add the corresponding edges to the graph if necessary before the message passing. This is visualized for the matrix element $A_{33}$ in Figure 1.

However, it is still possible that the output of the GNN results in a singular matrix as the final edge representations, obtained from message passing, can result in zero diagonal elements. Through additional

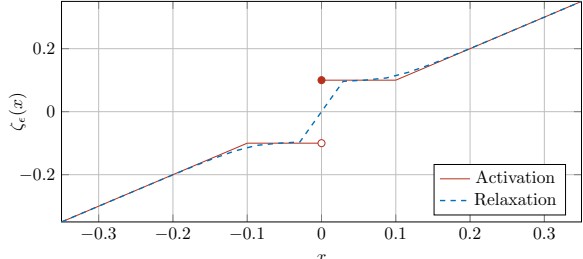

**Figure 2.** Plot of the activation function (3) for $\epsilon = 0.1$ and the corresponding relaxation (4).

measures, we avoid this in both factors and enforce non-singularity. Inspired by the classical LU factorization, we enforce a unit diagonal for the $\boldsymbol{U}$ factor. For the diagonal elements in the lower triangular factor $\boldsymbol{L}$ we are using the activation function

$$\zeta_\epsilon(x) = \begin{cases} x, & \text{if } |x| > \epsilon, \\ \epsilon, & \text{if } 0 \leq x \leq \epsilon, \\ -\epsilon, & \text{if } -\epsilon \leq x < 0. \end{cases} \quad (3)$$

In our experiments, we use the fixed hyper-parameter $\epsilon = 10^{-4}$. This guarantees that the output matrix $\boldsymbol{L}$ is invertible as diagonal elements with a small magnitude are shifted away from zero. However, this function is discontinuous at $x = 0$ making it difficult to optimize using gradient-based methods. Therefore, we use the following continuous approximation of the activation function:

$$\hat{\zeta}_\epsilon(x) = x \cdot \left(1 + \exp\left(-\left|\frac{4x}{\epsilon}\right| + 2\right)\right) \quad (4)$$

during training. The plots for the activation function and the proposed relaxation are shown in Figure 2.

## 3.2 Model training

The goal of the learned preconditioner $\boldsymbol{P} = \boldsymbol{LU}$ is to improve the spectral properties of the preconditioned linear equation system $\boldsymbol{AP}^{-1}$. However, as Sappl et al. [22] suggest, directly optimizing the system's condition number $\kappa$ becomes computationally intractable for large-scale problems. For computational tractability, we are also not taking the whole spectrum of the equation system into account but concentrating on the edges of the spectrum. In other words, we consider only the largest and smallest singular values of the preconditioned equation system.

**Lemma 1** *The largest singular value of the matrix $\boldsymbol{AP}^{-1}$ is upper bounded by the Frobenius norm:* $\sigma_{\max}(\boldsymbol{AP}^{-1}) \leq \epsilon^{-1}\|\boldsymbol{A} - \boldsymbol{P}\|_F + 1.$

Here, $\epsilon$ is the hyper-parameter of the preconditioner chosen in Equation (3). Further, we can estimate the Frobenius norm using Hutchinson's trace estimator [23] which allows us to express the loss to

minimize the upper bound on the largest singular value of the system in the following form:

$$\mathcal{L}_{\max}(\boldsymbol{P}; \boldsymbol{A}) = \|\boldsymbol{Aw} - \boldsymbol{Pw}\|_2^2, \quad (5)$$

where $\boldsymbol{w}$ is a standard normal distributed vector. This loss function has been previously applied by Häusner et al. [5] as an unbiased estimator of the squared Frobenius norm distance.

**Lemma 2** *The smallest singular value of $\boldsymbol{AP}^{-1}$ is lower bounded by the following inequalities:* $\sigma_{\min}(\boldsymbol{AP}^{-1}) \geq \|\boldsymbol{PA}^{-1}\|_F^{-1} \geq (\|\boldsymbol{PA}^{-1} - \boldsymbol{I}\|_F + 1)^{-1}.$

Based on this lemma, we obtain the following loss function[3] by using a similar approximation as in the previous step and optimizing over the inverse:

$$\mathcal{L}_{\min}(\boldsymbol{P}; \boldsymbol{A}) = \|\boldsymbol{PA}^{-1}\boldsymbol{w} - \boldsymbol{w}\|_2^2. \quad (6)$$

The drawback of this loss is that it requires computing $\boldsymbol{A}^{-1}\boldsymbol{w}$ during training. In other words, training the learned preconditioner requires solving many potentially ill-conditioned linear systems.

To maintain computational efficiency and avoid the need to solve equation systems online during training, we can instead utilize a supervised dataset composed of tuples $(\boldsymbol{A}_i, \boldsymbol{x}_i, \boldsymbol{b}_i)$, where each tuple satisfies $\boldsymbol{A}_i\boldsymbol{x}_i = \boldsymbol{b}_i$. This requires us to solve each training problem only once which can be computed beforehand in an offline fashion. We replace the standard distributed samples $\boldsymbol{w}$ in Equation (6) with the available training samples, resulting in a biased approximation of the original loss function:

$$\hat{\mathcal{L}}_{\min}(\boldsymbol{P}; \boldsymbol{A}) = \|\boldsymbol{PA}^{-1}\boldsymbol{b} - \boldsymbol{b}\|_2^2 = \|\boldsymbol{Px} - \boldsymbol{b}\|_2^2. \quad (7)$$

This loss function has been previously introduced by Li et al. [6] as an inductive bias introduced by the dataset for the Frobenius norm. Recently, Trifonov et al. [7] derived this loss function and the unbiased variant from Equation (6) based on reweighing the result from Lemma 1 using the inverse matrix as weights. They conjecture that optimizing this loss leads to minimizing the low-frequency components in the preconditioned equation system resulting in larger small singular values.

By combining the two previous loss functions from Equation (6) and Equation (5), we introduce a novel combined loss with the goal to further improve the conditioning by taking into account both ends of the spectrum. The combined loss function, along with its approximation using a supervised dataset similar to Equation (7), is defined as:

$$\begin{aligned}\mathcal{L}_{\text{com}}(\boldsymbol{P}; \boldsymbol{A}) &= \|\boldsymbol{Aw} - \boldsymbol{Pw}\|_2^2 + \alpha\|\boldsymbol{PA}^{-1}\boldsymbol{w}\|_2^2 \\ &\approx \|\boldsymbol{Aw} - \boldsymbol{Pw}\|_2^2 + \alpha\|\boldsymbol{Px}\|_2^2,\end{aligned} \quad (8)$$

where $\alpha$ is a hyper-parameter that controls the trade-off between the two loss components. During

---

[3]By convention, we always minimize the loss function.

training, we are minimizing the empirical risk over the corresponding unsupervised or supervised dataset to learn the parameters:

$$\hat{\boldsymbol{\theta}} \in \arg\min_{\boldsymbol{\theta}} \sum_{i=1}^{|\mathcal{D}|} \mathcal{L}(\boldsymbol{P_\theta}; \boldsymbol{A}_i, \boldsymbol{x}_i, \boldsymbol{b}_i). \qquad (9)$$

The proofs for the lemmas and the Froebnius norm approximation can be found in Appendix C.

# 4 Experiments & Results

We evaluate our method on a synthetic dataset of problems arising from the discretization of the Poisson equation. The problem size in our experiments is $n = 2\,500$. We train on 200 and evaluate the preconditioner on 10 unseen problems. It takes 5 minutes to train the model for 100 epochs excluding the time for the dataset generation. The implementation details can be found in Appendix B.

**Comparison of loss functions** In Table 1, we present the performance of the learned preconditioner given different loss functions corresponding to the bounds derived in the previous section. We can see that minimizing $\mathcal{L}_{\max}$ and $\mathcal{L}_{\min}$ decrease the largest and increase the smallest singular value of the system respectively and the models achieve the lowest loss for the respective functions. This shows that during training, we are able to minimize the loss functions and by that implicitly optimize the bounds for the singular values.

Optimizing the biased approximation $\hat{\mathcal{L}}_{\min}$ still performs well in terms of decreasing the original unbiased loss function. However, when computing the preconditioner inverse and singular value decomposition, numerical instabilities prevent the application of the learned output due to poor conditioning. This leads, in turn, to significantly worse performance in terms of GMRES iterations.

Optimizing the combined loss $\mathcal{L}_{\text{com}}$ reduces both individual loss functions slightly compared to the non-preconditioned case but leads to worse performance on nearly all metrics besides the acceleration of the iterative solver. Additionally, the training of this loss can be numerically unstable as the two loss functions conflict with each other.

Note that, the problem at hand is unbalanced as the spectrum of the original system is already skewed towards small singular values. In terms of GMRES iterations, the performance of the different loss functions also depends on the initial distribution of singular values. In Figure 3 we show this distribution for different preconditioners compared to the original distribution. Note that the original distribution is shifted to small singular values. We discuss the results in more detail in Appendix D.1.

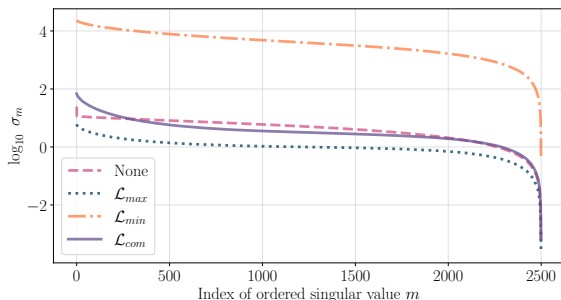

**Figure 3.** Descending log-scale plot of ordered singular values of a single problem instance for learned preconditioners derived from the different loss functions.

**Comparison with data-driven preconditioners** We additionally compare our method with previously developed data-driven preconditioners for incomplete Cholesky factorizations (Learned IC) with an exp-activation function for the diagonal elements to ensure invertibility. We use the loss function $\mathcal{L}_{\max}$ to train the method, similar to previous approaches [5], and choose the same hyper-parameters as our model for a fair comparison. The details of the baseline architecture are provided in Appendix B.3.

The results indicate that, although the Learned IC method reduces the number of iterations compared to the unpreconditioned GMRES solver, it performs worse in terms of both loss and conditioning when compared to our proposed method. Further, the GMRES performance is significantly worse than our proposed LU factorization-based techniques. This is, however, not very surprising as the LU factorization offers more flexibility to approximate the true matrix factorization as two separate factors are learned.

**Comparison with classical preconditioners** Additionally, we compare in Table 1 the performance of the learned preconditioner against the classical Jacobi and incomplete LU method without fill-ins, as well as the case where no preconditioner is applied. In general, the results show the vital importance of selecting an appropriate preconditioner. For the problem considered, ILU and the learned preconditioner perform nearly on par in terms of required iterations to solve the problem instances. We can see that even though the conditioning in terms of the largest and smallest singular value of the ILU method is by far inferior to the other preconditioners, it still manages to substantially reduce the number of iterations for convergence.

This shows that the spectrum edges, while important, are not the only factor affecting the method's efficiency. Further, this underscores the importance of considering the entire spectrum of the preconditioned system when analyzing the method's convergence as well as the alignment of the right-hand side $\boldsymbol{b}$ of the system with the singular vectors of $\boldsymbol{A}$ [3, 24].

**Table 1.** Average performance of classical and data-driven preconditioners with different loss functions evaluated on 10 test samples. We show the conditioning of the right-preconditioned system $\boldsymbol{AP}^{-1}$ in the first three columns indicating the smallest ($\sigma_{\min}$) and largest ($\sigma_{\max}$) singular values as well as the condition number ($\kappa$). For each preconditioner, we compute different Frobenius norm loss functions proportional to the bounds in Lemma 1 and Lemma 2. Further, we compare the convergence of the preconditioner in the GMRES algorithm in terms of the total time (including both the time to compute the preconditioner and solve the systems) and the number of iterations. For the unpreconditioned method, we compute the loss for $\boldsymbol{P} = \boldsymbol{I}$.

| | Method | $\sigma_{\min} \uparrow$ | $\sigma_{\max} \downarrow$ | $\kappa \downarrow$ | $\|\boldsymbol{P} - \boldsymbol{A}\|_F \downarrow$ | $\|\boldsymbol{PA}^{-1} - \boldsymbol{I}\|_F \downarrow$ | Time $\downarrow$ | Iterations $\downarrow$ |
|---|---|---|---|---|---|---|---|---|
| **Precond.** | No preconditioner | 0.0014 | 20.70 | 31 152.94 | 255.26 | 1 577.65 | 30.85 | 1 153 |
| | Jacobi | 0.0003 | 5.16 | 31 166.13 | 205.83 | 6 319.45 | 30.12 | 1 152 |
| | ILU(0) | 0.0006 | 30.32 | 120 740.90 | 138.31 | 3 688.45 | **3.33** | **413** |
| | Learned IC | 0.0006 | 7.58 | 27 405.57 | 143.23 | 3 719.47 | 12.84 | 692 |
| **Loss** | $\mathcal{L}_{\max}$: Equation (5) | 0.0007 | **5.00** | **16 139.46** | **88.76** | 3 261.23 | 3.67 | 437 |
| | $\mathcal{L}_{\min}$: Equation (6) | **1.1375** | 20 318.88 | 37 030.72 | 287.62 | **50.05** | 24.41 | 1 054 |
| | $\hat{\mathcal{L}}_{\min}$: Equation (7) | - | - | - | 287.71 | 50.30 | 130.01 | 2 192 |
| | $\mathcal{L}_{\text{com}}$: Equation (8) | 0.0017 | 52.92 | 66 691.71 | 197.82 | 1 240.60 | 3.42 | 418 |

# 5   Discussion & Conclusion

In this paper, we learn incomplete LU factorizations of sparse matrices directly from data using graph neural networks. When applied as a preconditioner our approach leads to a significant reduction in terms of iterations compared to the unpreconditioned GMRES and performs similar to classical incomplete factorization methods. Combining data-driven methods with existing numerical algorithms has a huge potential since it allows obtaining the best results from both worlds: guarantees about the convergence are obtained from classical analysis of the methods while the data-driven components allow us to learn directly from data to improve upon the existing algorithms on specific data distributions.

**Limitations** Our approach assumes having access to a distribution of linear equation systems that share similarities between the problem instances. However, in practice, it is not always trivial to obtain such a distribution of problems. Further, the initial cost of training the network needs to be amortized over many future problems that need to be solved. However, given the initial sample problems, new problems can be solved faster in an online setting which is important for time-critical applications.

In our experiments, we only train and evaluate our method on a single dataset of very limited size which is inspired by numerical computations. Applying the method to more diverse problem classes with different initial distributions of singular values is required in the future to obtain more general results about the performance.

Although the combined loss function $\mathcal{L}_{\text{com}}$ accelerates GMRES convergence, the conflicting loss terms can introduce instability during training. Additionally, finding an appropriate balance between the two components using the hyper-parameter $\alpha$ is challenging, as a poorly chosen value might emphasize one part of the spectrum more than intended, resulting

in a similar performance to using one of the loss functions individually.

**Future work** Our GNN is designed such that the learned preconditioner matches the sparsity pattern of the input matrix $\boldsymbol{A}$. While dynamically learning the sparsity pattern and matrix elements could improve preconditioner quality, the large search space and combinatorial complexity make this challenging.

Our proposed loss function accounts for the edges of the spectrum in the preconditioned system. However, beyond the edges and condition number, the distribution of singular values as well as the right-hand side also affects the convergence of Krylov subspace methods [24]. Designing loss functions that consider these additional aspects can further improve the performance of learned preconditioners.

An interesting idea for future work is to combine our data-driven incomplete LU factorization with classical ILU methods along the lines of the method proposed by Trifonov et al. [7] combining the advantages of classical and data-driven methods. The theoretical analysis of the different loss functions from our results can then be applied to construct novel hybrid preconditioners.

# Acknowledgments

We thank Daniel Hernández Escobar and Sebastian Mair for helpful discussions and feedback. This work was supported by the Göran Gustafsson Foundation and the Wallenberg AI, Autonomous Systems and Software Program (WASP) funded by the Knut and Alice Wallenberg Foundation.

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

# A  GMRES algorithm

The right-preconditioned GMRES is shown in Algorithm A.1. The non-preconditioned version can be obtained by using the identity matrix as a preconditioner ($\boldsymbol{P} = \boldsymbol{I}$). Operations where the preconditioner is applied are highlighted for better readability in this section.

**Algorithm A.1** RIGHT-PRECONDITIONED GMRES [1]

1: **Inputs:**
2: Non-singular matrix $\boldsymbol{A} \in \mathbb{R}^{n \times n}$
3: Preconditioner $\boldsymbol{P} \in \mathbb{R}^{n \times n}$
4: Right-hand side $\boldsymbol{b} \in \mathbb{R}^n$
5: Initial guess $\boldsymbol{x}_0 \in \mathbb{R}^n$
6: Tolerance $\epsilon$ for the residual norm
7: Maximum number of iterations $k_{\max}$
8: **Output:** Approximate solution $\boldsymbol{x}_k$
9: $k = 0$
10: $\triangleright$ *Compute the initial residual*
11: $\boldsymbol{r}_0 = (\boldsymbol{b} - \boldsymbol{A}\boldsymbol{x}_0)$
12: $\rho_0 = \|\boldsymbol{r}_0\|_2$
13: $\beta = \rho_0$
14: **while** $\rho_k > \epsilon\rho_0$ **and** $k < k_{\max}$ **do**
15:    $k \leftarrow k + 1$
16:    Arnoldi $(\boldsymbol{A}, \boldsymbol{P}, \boldsymbol{r}_0, k) \rightarrow \boldsymbol{V}_{k+1}, \boldsymbol{H}_{k+1,k}$
17:    $\triangleright$ *Compute the QR factorization of $\boldsymbol{H}_{k+1,k}$*
18:    $\rho_k = |\beta q_{1,k+1}|$
19:    $\triangleright$ *Solve using the QR factorization of $\boldsymbol{H}_{k+1,k}$*
20:    $\boldsymbol{y}_k = \underset{\boldsymbol{y}\in\mathbb{R}^k}{\arg\min} \|\beta\boldsymbol{e}_1 - \boldsymbol{H}_{k+1,k}\boldsymbol{y}\|_2$
21:    $\triangleright$ *Construct the solution*
22:    $\boldsymbol{x}_k = \boldsymbol{x}_0 + \boldsymbol{P}^{-1}\boldsymbol{V}_k\boldsymbol{y}_k$
23: **return** $\boldsymbol{x}_k$

**Arnoldi method** In each step of the iterative scheme (line 16 in Algorithm A.1), the Arnoldi method is used to construct an orthonormal basis of the Krylov subspace $\mathcal{K}_k$ of the current iteration shown in Equation (1). This method is crucial as it ensures that the basis vectors are orthogonal, which in turn allows the GMRES algorithm to efficiently minimize the residual over the Krylov subspace.

For a given number of iterations $k$, the Arnoldi method (Algorithm A.2) produces an upper Hessenberg matrix $\boldsymbol{H}_{k,k+1}$, which is related to the matrix $\boldsymbol{A}$ through the orthonormal basis $\boldsymbol{V}_{k+1}$ as

$$\boldsymbol{A}\boldsymbol{V}_k = \boldsymbol{V}_{k+1}\boldsymbol{H}_{k,k+1}, \tag{10}$$

where $\boldsymbol{V}_k$ consists of the first $k$ columns of the matrix $\boldsymbol{V}_{k+1}$, forming an orthonormal basis for the Krylov subspace.

Based on this factorization of this basis, the residual and approximate solution can be efficiently obtained by solving a smaller and upper-triangular system derived from the QR factorization of $\boldsymbol{H}_{k,k+1}$, as shown in line 20 of the GMRES algorithm.

In lines 21–23 of Algorithm A.2, the situation where the element $h_{k+1,k}$ in the upper Hessenberg matrix $\boldsymbol{H}_{k+1,k}$ becomes zero is addressed. If this condition is met, the loop breaks, indicating that the algorithm has prematurely found an invariant Krylov subspace. When this occurs, the $(k + 1)$-th column of $\boldsymbol{V}_{k+1}$ does not exist because the last row of $\boldsymbol{H}_{k+1,k}$ is zero. As a result, Equation (10) simplifies to

$$\boldsymbol{A}\boldsymbol{V}_k = \boldsymbol{V}_k\boldsymbol{H}_k.$$

This indicates that the Krylov subspace has reached its full dimension, and no further vectors can be generated. Consequently, the GMRES algorithm can terminate early, as the exact solution lies within the current subspace. This premature convergence is referred to as "lucky" because it implies that the solution has been found in less than $n$ iterations.

Also, note that in practice, it is not necessary to run the full Arnoldi algorithm in each iteration but the basis vectors found in the previous iterations can be reused and only the most recent orthonormalization step needs to be executed.

**Algorithm A.2** ARNOLDI'S MODIFIED GRAM-SCHMIDT IMPLEMENTATION

1: **Inputs:**
2: Non-singular matrix $\boldsymbol{A} \in \mathbb{R}^{n \times n}$
3: Preconditioner $\boldsymbol{P} \in \mathbb{R}^{n \times n}$
4: Initial residual $\boldsymbol{r}_0 \in \mathbb{R}^n$
5: Number of iterations $k$
6: **Outputs:**
7: Orthonormal basis $\boldsymbol{V}_{k+1} = \{\boldsymbol{v}_j\}_{j=1}^{k+1}, \boldsymbol{v}_j \in \mathbb{R}^n$
8: Hessenberg matrix $\boldsymbol{H}_{k+1,k} \in \mathbb{R}^{(k+1)\times k}$
9: $\triangleright$ *Initialize the Arnoldi basis*
10: $\beta = \|\boldsymbol{r}^{(0)}\|_2$
11: $\boldsymbol{v}_1 = \beta^{-1}\boldsymbol{r}^{(0)}$
12: **for** $i = 1$ to $k$ **do**
13:    $\boldsymbol{w}_{i+1} = \boldsymbol{A}\boldsymbol{P}^{-1}\boldsymbol{v}_i$
14:    $\triangleright$ *Gram-Schmidt ortho-normalization*
15:    **for** $j = 1$ to $i$ **do**
16:       $h_{j,i} = \boldsymbol{w}_{i+1}^{\mathsf{T}}\boldsymbol{v}_j$
17:       $\boldsymbol{w}_{i+1} = \boldsymbol{w}_{i+1} - h_{j,i}\boldsymbol{v}_j$
18:    $h_{i+1,i} = \|\boldsymbol{w}_{i+1}\|_2$
19:    **if** $h_{i+1,i} \neq 0$ **then**
20:       $\boldsymbol{v}_{i+1} = \boldsymbol{w}_{i+1}/h_{i+1,i}$
21:    **else**
22:       $\triangleright$ *Exact solution found (lucky breakdown)*
23:       **break**
24: **return** $\boldsymbol{V}_{k+1}, \boldsymbol{H}_{k+1,k}$

**Preconditioning** The preconditioning matrix $\boldsymbol{P}$ (or its inverse) does not need to be stored explicitly. Instead, it is sufficient to have access to an implicit representation of the matrix which can be applied to a vector $\boldsymbol{v}$ to evaluate the product $\boldsymbol{P}^{-1}\boldsymbol{v}$ as in line 22 of Algorithm A.1 and Algorithm A.2 respectively.

In the case of preconditioner given by triangular factorizations such as ILU and our learned approach, the matrix is obtained by solving the system $\boldsymbol{LUv} = \boldsymbol{r}$ which can be achieved in $\mathcal{O}(n^2)$ operations using the forward-backward substitution and can be further accelerated by exploiting the sparse structure of the matrices. The solution to solving these systems is equivalent to explicitly inverting the matrices and computing the matrix-vector product $\boldsymbol{U}^{-1}\boldsymbol{L}^{-1}\boldsymbol{r}$ which forms the preconditioning matrix as $\boldsymbol{P}^{-1} = (\boldsymbol{LU})^{-1}$. However, solving the triangular systems is more efficient in practice as the generally non-sparse inverse matrices do not need to be stored.

**Convergence** Obtaining descriptive convergence bound is in general non-trivial due to the complex nature of GMRES. For non-symmetric matrices, singular values provide a more complete understanding of a matrix's behavior across all directions in space compared to eigenvalues. Eigenvalues are tied to specific directions – eigenvectors – that remain unchanged under transformation by the matrix $\boldsymbol{A}$. However, non-symmetric matrices may lack sufficient eigenvectors to describe their impact across all directions, particularly if they are defective or not diagonalizable. In contrast, singular values offer a full directional picture. The matrix can be decomposed, using singular value decomposition (SVD), as $\boldsymbol{A} = \boldsymbol{U\Sigma V}^{\mathsf{T}}$, where $\boldsymbol{\Sigma}$ is a diagonal matrix of singular values, and $\boldsymbol{U}$ and $\boldsymbol{V}$ are orthogonal matrices containing left and right singular vectors, respectively. These singular vectors capture $\boldsymbol{A}$'s scaling effects across all directions, making singular values especially valuable for analyzing convergence and stability in iterative methods that require complete directional information [1, 12]. As the generated Krylov subspace in GMRES highly depends on the matrix's scaling behavior, SVD offers the tools to understand the iterative process through the transformations of $\boldsymbol{A}$ [25].

The condition number, $\kappa(\boldsymbol{A}) = \frac{\sigma_{\max}(\boldsymbol{A})}{\sigma_{\min}(\boldsymbol{A})}$, defined by the largest and smallest singular values of $\boldsymbol{A}$ is a popular metric to assess the convergence of iterative methods. This metric describes how well-conditioned the matrix is, with a lower $\kappa(\boldsymbol{A})$ indicating better clustering of singular values, typically around 1 in well-conditioned systems. A smaller $\kappa(\boldsymbol{A})$ often leads to faster convergence since all directions contribute meaningfully to residual reduction. However, if $\kappa(\boldsymbol{A})$ is large due to a very small $\sigma_{\min}(\boldsymbol{A})$, convergence slows as the matrix has limited influence in the directions corresponding to small singular values. This condition can cause stagnation, requiring more iterations to meet convergence criteria as $\kappa(\boldsymbol{A})$ grows [1, 12].

Preconditioning addresses this by adjusting the singular value spectrum, ideally reducing $\kappa(\boldsymbol{A})$ to cluster singular values more effectively and ensuring significant influence in all directions of the subspace.

A well-chosen preconditioner modifies $\boldsymbol{A}$'s singular values to improve conditioning, thus lowering the number of iterations needed to achieve a desired tolerance. This approach makes GMRES particularly effective for systems that are well-conditioned or well-preconditioned, solidifying singular values as the preferred metric over eigenvalues for evaluating convergence and stability in this context [3].

**Right-hand side** Apart from the condition number and singular value distribution, the right-hand side $\boldsymbol{b}$ in the system also plays a crucial role in GMRES convergence through its interaction with the singular value decomposition of $\boldsymbol{A}$ or the preconditioned system. Decomposing $\boldsymbol{b} = \sum_{i=1}^{n} \alpha_i \boldsymbol{u}_i$, with $\{\boldsymbol{u}_i\}$ as the left singular vectors of $\boldsymbol{A}$, reveals how $\boldsymbol{b}$ aligns with $\boldsymbol{A}$'s left-singular values. The coefficients $\alpha_i = \boldsymbol{u}_i^{\mathsf{T}} \boldsymbol{b}$ measure $\boldsymbol{b}$'s alignment with each left singular vector $\boldsymbol{u}_i$. Strong alignment with directions tied to small singular values $\sigma_i$ causes these directions to dominate the residual and slow down GMRES convergence [3, 12].

Our data-driven preconditioning method indirectly addresses the interaction between the right-hand side $\boldsymbol{b}$ and the singular values of $\boldsymbol{A}$ by sampling $\boldsymbol{b}$ from a distribution in the approximate loss functions. This ensures robustness by forcing the preconditioner to handle diverse right-hand sides and target weakly scaled directions of $\boldsymbol{A}$. However, our method does not explicitly exploit the alignment of a specific right-hand side $\boldsymbol{b}$ with the singular vectors of $\boldsymbol{A}$. To refine this approach, the loss function can be enhanced in future work to incorporate the dependence of a specific right-hand side and its alignment with the singular values explicitly.

# B    Implementation details

Here, we provide additional details for the implementation of the dataset, our learned preconditioner, and the baseline preconditioners used in the experiments.

## B.1    Dataset

Our goal with the provided dataset is not to solve a real-world problem. Rather, we focus on a highly ill-conditioned synthetic dataset which allows us to systematically test the different loss functions derived. For the problem scale used, direct methods are much superior compared to the shown results.

**Poisson problem** The Poisson equation is an elliptic partial differential equation (PDE) and one of the most fundamental problems in numerical computational science [26]. The general form of the Poisson equation is given by:

$$-\nabla^2 u(x) = f(x) \qquad x \in \Omega, \qquad (11)$$
$$u(x) = u_D(x) \qquad x \in \partial\Omega, \qquad (12)$$

where $f(x)$ is the source function and $u(x)$ is the unknown function to be solved for.

In our study, we generate the matrix $\boldsymbol{A}$ using `pyamg.gallery.poisson`, which implicitly assumes unit spacing and Dirichlet boundary conditions, typically set to zero. The right-hand side of the linear equation system, $\boldsymbol{b}$, is derived from the source function $f(x, y) = \sin(\pi x) \sin(\pi y)$. By discretizing the problem using the finite element method, we obtain a system of linear equations of the form $\boldsymbol{Ax} = \boldsymbol{b}$, where the stiffness matrix $\boldsymbol{A}$ is sparse and symmetric positive definite. Throughout our experiments, we use matrices of size $n = 2\,500$.

**Noisy data** However, we are not specifically interested in symmetric and positive definite matrices that are generated through finite element discretization. Therefore, to obtain a distribution $\mathbb{A}$ over a large number of Poisson equation PDE problems with more general structure that can be efficiently solved using GMRES, we perturb all the non-zero entries with standard Gaussian noise, which yields arbitrary matrices not necessarily spd. Let us denote the perturbed matrix as $\boldsymbol{A}' = (a'_{ij})$. We can represent this mathematically by:

$$a'_{ij} = \begin{cases} a_{ij} + X_{ij} & \text{if } a_{ij} \neq 0, \\ 0 & \text{if } a_{ij} = 0, \end{cases}$$

where $X_{ij} \sim \mathcal{N}(0, 1)$ are i.i.d. random variables.

By introducing randomness into $\boldsymbol{A}$, we are effectively modeling a scenario where the system's properties are not perfectly regular, which could represent, for example, some form of physical or numerical irregularity or perturbation in the grid or medium. During testing, we use a deterministic right-hand side $\boldsymbol{b}$ as described previously. However, for the supervised training dataset we sample the vectors $\boldsymbol{b}$ to be normally distributed in order to avoid overfitting as this would lead to a significant decrease in model performance.

**Problem instances** All problem instances generated come from the same distribution but using non-overlapping random seeds we ensure that there is no data leakage between the training and test data. We create 200 problem instances for training and 10 instances for validation and testing respectively.

All problems are of size $n = 2\,500$ with approximately $50\,000 - 150\,000$ non-zero elements. The condition number $\kappa$ of the unpreconditioned system is in the range between $5\,000$ to $50\,000$. Thus, even though the problem parameters only vary slightly, the resulting conditioning and potential performance of the different loss functions can be very different between the individual problem instances.

## B.2 Network architecture

We implement the neural network based on the `pytorch-geometric` framework which offers direct support for computing the node features and utility functions such as adding remaining self-loops [27]. During inference, we utilize the `numml` sparse matrix package to efficiently compute the forward-backward substitution [28]. The pseudocode for the forward pass of the learned preconditioner is shown in Algorithm B.1.

**Inputs and transformations** As described in Section 3, we transform the original linear equation system matrix $\boldsymbol{A}$ into a graph $\mathcal{G}$ using the Coates graph representation. We use the non-zero elements of the matrix as edge features and add a second edge feature as a positional encoding of the output. For the node features, we utilize the same set of 8 input features as previously applied by Häusner et al. [5] describing the structural properties of the matrix:

- the node degree $\deg(v)$
- the maximum degree of neighboring nodes
- the minimum degree of neighboring nodes
- the mean degree of neighboring nodes
- the variance of degrees of neighboring nodes
- the diagonal dominance
- the diagonal decay
- the position of the node in the linear system

**Message-passing network** In our implementation, we use $L = 3$ message-passing steps. We define a hidden size of $n = 32$ for the edge embeddings and $m = 16$ for the node embeddings in the hidden layers of the neural network allowing for sufficient flexibility in the network's parameterization. The incoming messages at each node are aggregated using the mean aggregation function, which ensures a balanced contribution from neighboring nodes. Thus, our GNN has a total of $4\,889$ parameters to train.

**Output** The final edge embedding is chosen to be scalar by designing the edge update function to produce a single output, i.e., $e_{ij}^{(L)} \in \mathbb{R}$, allowing us to directly transform the output into a lower- and upper-triangular matrix. We enforce the diagonal elements of $\boldsymbol{U}$ to be ones and apply the activation function $\zeta$ from Equation (3) to the diagonal elements of $\boldsymbol{L}$. Note that during training, we instead use the approximation $\hat{\zeta}_\epsilon$ from Equation (4) in line 25 as discussed in the main text. This allows zero diagonal elements but no matrix inversion of $\boldsymbol{P}$ during training is required.

**Algorithm B.1** PSEUDO-CODE FOR THE LEARNED LU FACTORIZATION PRECONDITIONER

1: **Input:** Coates graph representation $\mathcal{G} = (\mathcal{V}, \mathcal{E})$ of the system matrix $\boldsymbol{A}$.
2: **Output:** Lower and upper triangular matrices $\boldsymbol{L}$ and $\boldsymbol{U}$ of the learned factorization.
3: ▷ *Input transformations:*
4: Add remaining self-loops to the graph $\mathcal{G}$.
5: Compute node features $\boldsymbol{n}_i \in \mathbb{R}^8$.
6: Compute edge features $\boldsymbol{e}_{ij} \in \mathbb{R}^2$.
7: ▷ *Message passing layers*
8: **for** $l \in \{0, 1, \dots, L-1\}$ **do**
9:    **for** $(i,j) \in \mathcal{E}$ **do**
10:      ▷ *Compute updated edge features*
11:      $\boldsymbol{e}_{ij}^{l+1} = \psi_{\boldsymbol{\theta}}(\boldsymbol{e}_{ij}^l, \boldsymbol{n}_i^l, \boldsymbol{n}_j^l)$
12:    **for** $i \in \{1, \dots, n\}$ **do**
13:      ▷ *Aggregate edge features per node*
14:      $\boldsymbol{m}_i^{l+1} = \bigoplus_{j \in \mathcal{N}(i)} \boldsymbol{e}_{ji}^{l+1}$
15:      ▷ *Compute updated node features*
16:      $\boldsymbol{n}_i^{l+1} = \phi_{\boldsymbol{\theta}}(\boldsymbol{n}_i^k, \boldsymbol{m}_i^{l+1})$
17:    **if** not final layer in the network **then**
18:      ▷ *Add skip connections*
19:      **for** $(i,j) \in \mathcal{E}$ **do**
20:      $\boldsymbol{e}_{ij}^{l+1} \leftarrow [\boldsymbol{e}_{ij}^{l+1}, a_{ij}]^\top$
21: ▷ *Form the lower triangular matrix $\boldsymbol{L} = (l_{ij})$:*

$$(l_{ij}) \leftarrow \begin{cases} e_{ij}^{(L)}, & \text{if } (i,j) \in \mathcal{E} \text{ and } i > j, \\ 0, & \text{otherwise.} \end{cases}$$

22: ▷ *Form the upper triangular matrix $\boldsymbol{U} = (u_{ij})$:*

$$(u_{ij}) \leftarrow \begin{cases} e_{ij}^{(L)}, & \text{if } (i,j) \in \mathcal{E} \text{ and } i < j, \\ 0, & \text{otherwise.} \end{cases}$$

23: ▷ *Ensure invertibility*
24: **for** $i \in \{1, \dots, n\}$ **do**
25:    $l_{ii} \leftarrow \zeta(e_{ii}^{(L)})$
26:    $u_{ii} \leftarrow 1$
27: **return** $(\boldsymbol{L}, \boldsymbol{U})$

**Training** Our model is trained on the dataset of 200 problem instances of size $n = 2\,500$ for 100 iterations. The best-performing model during the training based on the validation GMRES performance is chosen in order to avoid overfitting. As a validation metric, we use the number of GMRES iterations of the learned preconditioner on the validation data as this is the downstream metric of interest.

We use an NVIDIA-A100 GPU with 80 GB of memory for training. Each epoch, consisting of 200 parameter update steps using the Adam optimizer with a learning rate of 0.001 and a batch size of 1, requires 2 seconds. Thus, the total time required for model training is approximately 5 minutes. Further during training we use gradient clipping with parameter 1. We assume throughout that a supervised dataset is readily available and we do not need to

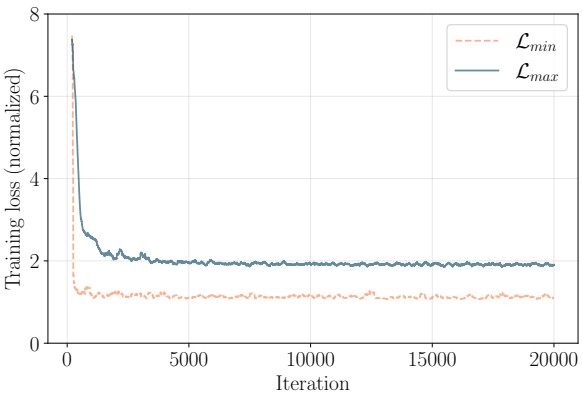

**Figure B.1.** Normalized training loss using different loss functions from Section 3.

create samples for the loss function Equation (7) before the training process. We use the right-hand sides $\boldsymbol{b}$ are generated as described in Section B.1.

For loss functions that require solving the system $\boldsymbol{A}^{-1}\boldsymbol{w}$, training takes significantly longer with approximately 20 minutes. However, solving the systems can be implemented efficiently for the problem scale that we consider in our experiments. For real-world examples, the additional time required would increase significantly.

**Tuning** We only conducted a minimal level of hyperparameter tuning for the model architecture. Tuning the loss function in Equation (8), optimization of the objective becomes numerically more difficult as the two terms are in conflict with each other. We choose the hyper-parameter $\alpha = {}^1/7$ based on the validation set performance to balance out the two loss terms of the combined loss.

## B.3 Baselines

The GMRES algorithm and the Arnoldi method, given in Algorithm A.1 and Algorithm A.2, respectively, are implemented in `pytorch` directly. We utilize an iterative implementation of the Arnoldi method that only computes one orthogonal vector in each iteration for computational efficiency. As the considered problems here are of moderate size, we apply a direct QR factorization to the matrix $\boldsymbol{H}$ at each step, rather than incrementally updating the factorization through Givens rotations. Although Givens rotations are beneficial for larger or sparser systems due to their ability to maintain efficiency in incremental updates, a direct QR factorization is computationally feasible here and simplifies implementation [12].

**Classical Preconditioner** The Jacobi preconditioner can be implemented and applied very efficiently using `pytorch` sparse data structures for diagonal matrices. For the ILU(0) preconditioner,

we use the `ILU++` implementation which is a high-performance implementation of the algorithm in C++ [29, 30].

**Data-driven preconditioner** The data-driven baseline uses an architecture similar to the one proposed by Li et al. [6]. The architecture by Häusner et al. [5] is not directly applicable since it only takes the lower triangular part of the matrix as an input which would remove many input features as the matrices in our dataset are non-symmetric. However, we use a exp-activation function for the diagonal elements instead of choosing the original elements in $\boldsymbol{A}$ since the matrices in our dataset are not guaranteed to have non-zero diagonal elements. Further, we use the same node and edge features as our GNN described in the previous section.

**Inference** During inference, we run all operations – including the neural network-driven preconditioner – on the CPU in order to ensure a fair comparison between the different methods. For all methods, the time it takes to compute the preconditioner can be neglected compared to the GMRES time.

# C  Proofs

In this section, we prove the lemmas from the main text. The goal of Lemmas 1 and 2 is to optimize the system's spectral properties by minimizing the upper and maximizing the lower spectral bounds respectively. This approach allows our loss function to increase the smaller singular values of the preconditioned system while decreasing the larger singular values, thereby improving the clustering of the singular values in the preconditioned matrix. The proofs can easily be verified and rely on basic norm transformations and norm inequalities.

**Proof of Lemma 1** In the first lemma, we aim to upper bound the largest singular value of the preconditioned linear system. The bound is derived as follows:

$$
\begin{aligned}
\sigma_{\max}(\boldsymbol{A}\boldsymbol{P}^{-1}) &= \|\boldsymbol{A}\boldsymbol{P}^{-1}\|_2 \\
&= \|(\boldsymbol{A} - \boldsymbol{P} + \boldsymbol{P})\boldsymbol{P}^{-1}\|_2 \\
&= \|(\boldsymbol{A} - \boldsymbol{P})\boldsymbol{P}^{-1} + \boldsymbol{I}\|_2 \\
&\leq \|(\boldsymbol{A} - \boldsymbol{P})\boldsymbol{P}^{-1}\|_2 + \|\boldsymbol{I}\|_2 \\
&\leq \|\boldsymbol{A} - \boldsymbol{P}\|_2 \, \|\boldsymbol{P}^{-1}\|_2 + 1.
\end{aligned}
$$

Now, we exploit the structure of the learned preconditioner to obtain an upper bound for $\|\boldsymbol{P}^{-1}\|_2$. Specifically, we use two key observations. First, the matrix $\boldsymbol{U}$ has ones on the diagonal by construction. Second, the absolute values of the diagonal elements on the matrix $\boldsymbol{L}$ are bounded from below by $\epsilon$, a property that arises from the specific choice of activation function in Equation (3).

These properties, when combined with the fact that the singular values of a triangular matrix coincide with the absolute value of its diagonal elements, allow us to derive the following upper bound on the spectral norm of the preconditioner $\boldsymbol{P}$:

$$
\begin{aligned}
\|\boldsymbol{P}^{-1}\|_2 \\
= \|\boldsymbol{U}^{-1}\boldsymbol{L}^{-1}\|_2 \\
\leq \|\boldsymbol{U}^{-1}\|_2 \, \|\boldsymbol{L}^{-1}\|_2 \\
= \frac{1}{\sigma_{\min}(\boldsymbol{U})} \cdot \frac{1}{\sigma_{\min}(\boldsymbol{L})} \\
\leq \epsilon^{-1}.
\end{aligned}
$$

Combining this bound on $\|\boldsymbol{P}^{-1}\|_2$ with the previous inequality, we obtain:

$$
\sigma_{\max}(\boldsymbol{A}\boldsymbol{P}^{-1}) \leq \epsilon^{-1}\|\boldsymbol{A} - \boldsymbol{P}\|_2 + 1.
$$

**Proof of Lemma 2** In the second lemma, we show how to obtain a lower bound on the smallest singular value of the preconditioned system and it is derived as follows:

$$
\begin{aligned}
\sigma_{\min}(\boldsymbol{A}\boldsymbol{P}^{-1}) &= \frac{1}{\sigma_{\max}(\boldsymbol{P}\boldsymbol{A}^{-1})} \\
&= \frac{1}{\|\boldsymbol{P}\boldsymbol{A}^{-1}\|_2} \\
&\geq \frac{1}{\|\boldsymbol{P}\boldsymbol{A}^{-1}\|_F}.
\end{aligned}
$$

Previous learned preconditioner approaches in the literature [6, 7] used the following additional steps to obtain a weaker bound on the smallest singular value of the system:

$$
\begin{aligned}
\|\boldsymbol{P}\boldsymbol{A}^{-1}\|_2 &= \|\boldsymbol{P}\boldsymbol{A}^{-1}\|_2 - 1 + 1 \\
&= \|\boldsymbol{P}\boldsymbol{A}^{-1}\|_2 - \|\boldsymbol{I}\|_2 + 1 \\
&\leq \big|\|\boldsymbol{P}\boldsymbol{A}^{-1}\|_2 - \|\boldsymbol{I}\|_2\big| + 1 \\
&\leq \|\boldsymbol{P}\boldsymbol{A}^{-1} - \boldsymbol{I}\|_2 + 1 \\
&\leq \|\boldsymbol{P}\boldsymbol{A}^{-1} - \boldsymbol{I}\|_F + 1.
\end{aligned}
$$

The reason for working with this second weaker approximation is that optimizing the first approximation directly leads to degenerate solutions as the minimum is attained when $\boldsymbol{P} \approx 0$ which would avoid having small singular values but the large singular value of the preconditioned system would be unbounded. Therefore, one can see the additional term involving $\boldsymbol{I}$ as a regularizer enforcing $\boldsymbol{P}$ to be non-degenerate.

**Frobenius norm approximation** Equivalently to optimizing over the Frobenius norm, we can optimize the preconditioner over the squared Frobenius norm instead as we are only interested in the best parameters, not the best objective value. This has the additional advantage that the loss function is fully differentiable. Further, we can approximate the

squared Frobenius norm using Hutchinson's trace estimator which can be easily verified [23]. For a random vector $\boldsymbol{w}$ that satisfies $\mathbb{E}\left[\boldsymbol{w}\boldsymbol{w}^{\mathsf{T}}\right] = \boldsymbol{I}$ we can approximate the squared Frobenius norm using the following transformations:

$$
\begin{aligned}
\|\boldsymbol{M}\|_F^2 &= \operatorname{trace}(\boldsymbol{M}^{\mathsf{T}}\boldsymbol{M}) \\
&= \operatorname{trace}(\boldsymbol{M}^{\mathsf{T}}\boldsymbol{M}\,\mathbb{E}\left[\boldsymbol{w}\boldsymbol{w}^{\mathsf{T}}\right]) \\
&= \mathbb{E}\left[\operatorname{trace}(\boldsymbol{M}^{\mathsf{T}}\boldsymbol{M}\boldsymbol{w}\boldsymbol{w}^{\mathsf{T}})\right] \\
&= \mathbb{E}\left[\operatorname{trace}(\boldsymbol{w}^{\mathsf{T}}\boldsymbol{M}^{\mathsf{T}}\boldsymbol{M}\boldsymbol{w})\right] \\
&= \mathbb{E}\left[\boldsymbol{w}^{\mathsf{T}}\boldsymbol{M}^{\mathsf{T}}\boldsymbol{M}\boldsymbol{w}\right] \\
&\approx \boldsymbol{w}^{\mathsf{T}}\boldsymbol{M}^{\mathsf{T}}\boldsymbol{M}\boldsymbol{w} \\
&= \|\boldsymbol{M}\boldsymbol{w}\|_2^2.
\end{aligned}
$$

In practice, $\boldsymbol{w}$ is typically chosen to follow either a standard normal distribution or a Rademacher distribution.

While it is possible to use multiple samples to obtain a more accurate approximation of the Frobenius norm, we follow previous approaches by estimating the norm using a single random vector to limit computational resources to create the dataset.

# D    Additional results

Here, we present additional results for the singular values, choice of activation function, and robustness with respect to different hyper-parameters $\epsilon$.

## D.1    Singular value distribution

We provide additional results for the baseline and learned preconditioners. In Figure D.1 we present the descending log-scale plot of ordered singular values for the baseline preconditioners, illustrating the same problem previously depicted in Figure 3. We can see that even though the ILU preconditioner has a few very large singular values, most values are clustered around 1 (Observe that the plot shows the logarithmic singular values thus the cluster appears

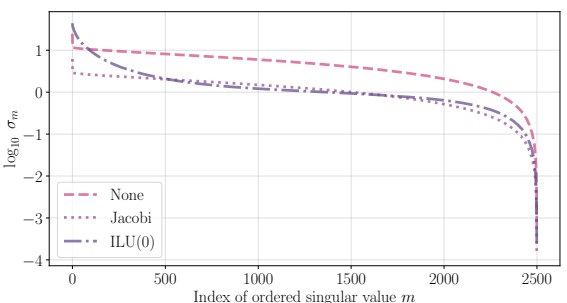

**Figure D.1.** Descending log-scale plot of ordered singular values of single problem instance for different baseline preconditioners.

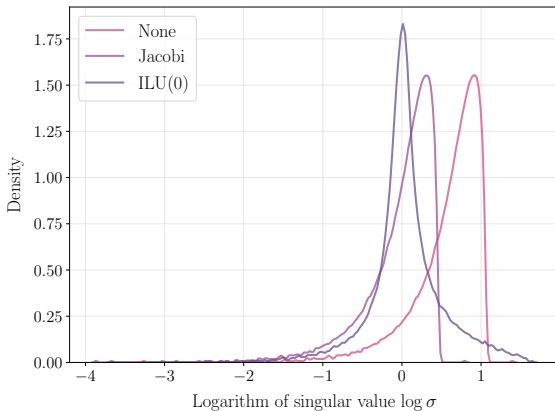

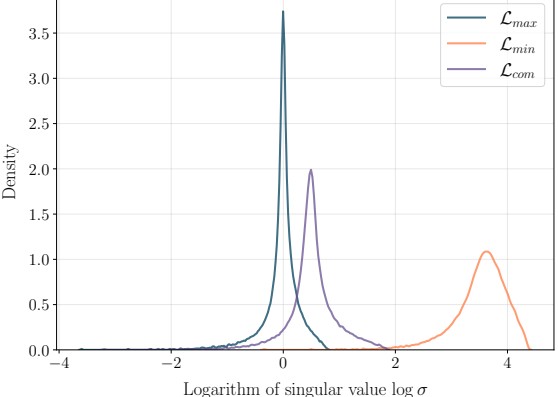

**Figure D.2.** Density plot showing the distribution of singular values for a single problem instance across various preconditioners.

at $0 = \log 1$). This leads to a very fast convergence as observed in Table 1 even though the condition number $\kappa$ of the problem is very high.

This fact gets even clearer when taking a look at the distribution of singular values shown in Figure D.2 for both the learned preconditioner and the baseline methods. We can see that the Jacobi preconditioner does not change the distribution of singular values significantly but only shifts the spectrum towards 1. Therefore, no improved convergence can be observed. The incomplete LU method leads to a tighter clustering of the singular values with a fast decay towards small singular values while some large singular values remain. For the learned methods, we can see that using the loss $\mathcal{L}_{\min}$ from Equation (6) leads to a distribution far away from zero but the singular values are not very clustered. The best clustering is achieved using the loss corresponding to large singular values $\mathcal{L}_{\max}$. However, the spectrum does not show a fast decay towards small singular values. In the spectrum of the combined loss, we can see the combined effect of the two approaches: the spectrum is clustered around a single value and the singular values are away from zero. These two properties of the spectrum make the preconditioned

system easier to solve leading to better performance observed in the experiments presented in Table 1 in the main text.

## D.2 Activation function

In order to evaluate the chosen activation function to ensure invertability, we compare the performance of the learned LU preconditioner with different values of the hyper-parameter $\epsilon$. Note that, the performance of the loss function and the chosen hyper-parameter also highly depends on the initial singular value distribution which is shifted to small singular values in our experiments as discussed previously.

The results of the learned preconditioner for 4 different values of $\epsilon$ are shown in Table D.1. We use the combined loss function $\mathcal{L}_{\text{com}}$ for all experiments. We can see that with a larger $\epsilon$ parameter, both the small singular values of the preconditioned system as well as the large singular values decrease. The best performance is achieved when balancing the two different ends of the singular value spectrum.

Finally, we can see that all models trained with the combined loss outperform the previous models trained on only one loss function showing the robustness of our proposed method.

**Connection to other data-driven techniques** Noteworthy, the method proposed by Li et al. [6] employs the loss function $\mathcal{L}_{\text{min}}$, which leads to better results compared to those observed in our experiments. However, a crucial difference between this method and our approach is the fact that the diagonal elements are not learned in the NeuralPCG preconditioner. Instead, the square root of the original elements of the matrix $\boldsymbol{A}$ is used. Since the matrices considered are positive definite, the diagonal elements are non-zero, and therefore, the incomplete factorization is guaranteed to be invertible. For general invertible matrices, that we explore in this paper, this is, however, not the case. The theoretical results presented here explain some of the discrepancies between our results and the previously obtained improvements. Implicitly, Li et al. [6] choose a large $\epsilon$ for these matrices by construction of the diagonal elements. This allows them to bound the largest singular value under the assumption that the differ-

ence between the original matrix and the learned factorization is not too large by applying Lemma 1. Further, it is easy to integrate the method developed here with data-driven methods for incomplete Cholesky methods [5]. This can be achieved by adding the $\epsilon$ parameter directly to the diagonal elements in order to ensure a lower bound on the values.

**Table D.1.** Performance of our method for different values of $\epsilon$ controlling the activation function. The columns presented here are a subset of the columns shown in Table 1.

| $\epsilon$ | $\sigma_{\text{min}} \uparrow$ | $\sigma_{\text{max}} \downarrow$ | $\kappa \downarrow$ | Its. $\downarrow$ |
|---|---|---|---|---|
| 0.0001 | **0.0022** | 39.12 | 37 513 | 420 |
| 0.001 | 0.0015 | 35.17 | 49 723 | **418** |
| 0.01 | 0.0012 | 17.26 | 30 349 | 428 |
| 0.1 | 0.0010 | **12.39** | **24 702** | 424 |

