# OpenReview forum: "Learning incomplete factorization preconditioners for GMRES"
_NLDL.org/2025/Conference — NLDL 2025 Oral_

### Official Review · Reviewer_dXKW · 2024-10-12

**Confidence:** 4

**Summary:**

This paper explores using neural networks to improve solving linear system of equations. Specifically, the paper looks at the GMRES method and improving the Kyrlov subspaces using a preconditioner.

The LU factorization of the preconditioner is found using a GNN. The paper presents two loss functions and compares the method against baselines.

**Strengths:**

The idea to compute the LU factorization is new and relevant to the community. Prior work has looked at computing eigendecompositions however these reply on the matrices being dense and PSD.

Here the matrices being factored are sparse and we are obtaining sparse partial factorizations as well.

The paper is very well written.

**Weaknesses:**

I’m not sure if the method in the paper works as thought. The loss function that optimizes for the top singular results in very large conditions numbers.

The same is true for the method that optimizes for both the max and min singular values. The iterations to convergence is reduced but it is unclear why this occurs.

The GNN used is not clear. What are the node features? These are not defined. Additionally the GNN only passes messages on the edges and not the nodes. This is also quite non standard.

**Justification:**

The paper presented a new method an interesting idea. The method seems to have reasonable properties and the idea I think can be of interest.

---

> ### Author Rebuttal · Authors · 2024-10-22
>
> We would like to thank the reviewer for their feedback and positive evaluation of our paper.
>
> > I’m not sure if the method in the paper works as thought.
>
> We have noticed that in Table 1 the results of singular values and condition number for using the loss function $\hat{\mathcal{L}}_\min$ were incorrect due to numerical issues in the inversion of the preconditioner. Therefore, we removed these numbers in the updated version and adjusted the discussion in the main text.
>
> Further, we added new figures and a discussion to Appendix D.1 where we discuss the here outlined questions. In summary, the convergence of GMRES depends on both the clustering of the singular values and the decay away from zero. The best combination of this is achieved with our loss function in the experiments. We hope that the additional discussion makes our suggested method more clear and thank the reviewer for the suggestion to clarify the reasons behinf the increase in performance.
>
> > The GNN used is not clear. What are the node features? These are not defined. Additionally the GNN only passes messages on the edges and not the nodes. This is also quite non standard.
>
> The node features are mentioned in the appendix only. We clarified the corresponding section and explicitly mention the used node features for our GNN. Apart from this, we use the GNN architecture presented in Section 2 which is a generic way to construct message-passing GNNs and covers many of the classical GNN architectures such as GCNs and GATs. The messages here are computed as an aggregation of the edge embeddings which are themselves computed as functions of the adjacent nodes. We hope that this clarifies the proposed GNN architecture.

---

### Official Review · Reviewer_3QJn · 2024-10-12
**Interesting paper!**

**Confidence:** 4

**Summary:**

The paper proposes an approach to replace hand-engineered algorithms for incomplete factorizations with a data-driven model based on GNNs that can be trained to optimize the preconditioner for specific problem distributions, leading to faster and more reliable performance in solving linear systems. In particular,  they use GNN model that learns and outputs incomplete LU factors for sparse matrices, providing a non-singular precondition. The work is aligned with using GNNs to solve linear algebra problems, as described in this paper: https://arxiv.org/abs/2310.14084.

One of the main contributions is the derivation of a new loss function that accounts for both large and small singular values of the system, improving the spectral properties and accelerating the GMRES iterations. Also, a particular activation function to ensure invertibility.

**Strengths:**

The paper is well written, the contribution seems valuable, and the problem is interesting.

**Weaknesses:**

The only downside is the experimental evaluation that is a bit limited and might benefit from an ablation study, e.g., to compare the proposed activation function with existing ones. Nevertheless, I believe the paper is worth to be published.

**Justification:**

Pros outweigh the downsides

---

> ### Author Rebuttal · Authors · 2024-10-22
>
> We thank the reviewer for the positive evaluation of the paper. We added additional experiments in the main text and in Appendix D.2 to improve the evaluation of our method.

---

### Official Review · Reviewer_NxYE · 2024-10-14
**Interesting data-driven approach to lower-upper factorisation of large-scale sparse matrices using graph neural networks**

**Confidence:** 3

**Summary:**

This manuscript introduces a data-driven approach to perform lower-upper (LU) factorisation of large-scale sparse matrices. The starting point is the generalised minimal residual method (GMRES) algorithm, which is one of the most popular methods for solving the task of LU factorisation in the case of large-scale spare matrices. This task is typically computationally demanding, and iterative methods are necessary. These iterative methods are highly dependent on the preconditioning methods, which are hyperparameter sensitive and can have difficulties converging. Therefore, a graph neural network (GNN)-based approach is proposed to learn the preconditioner instead. By precomputing matrix factorisation to create a supervised dataset, the GNN can learn to perform the task and solve new factorisation in less iterations. The writing is clear and the results are encouraging and well presented.

**Strengths:**

1. LU factorisation of large-scale sparse matrices is highly common in machine learning problems, and contributions toward making this process faster and more reliable could have a great impact in numerous branches of machine learning.

2. The use of GNNs seems like a suitable tool for the task, and loss functions follow nicely from established theory.

3. The writing is clear, the manuscript is well structured, and the math is presented in an understandable manner at a suitable detail level.

**Weaknesses:**

1. Other deep learning-based baselines could improve the impression of the results. The comparison with the classical preconditions is suitable and welcome, but it would also be useful to see how alternative deep learning-based approaches could work in this problem setting. For instance, the work of Chen [1] would be impractical due to the requirement of retraining for each problem, but would at least give an indication to what the performance of a different deep learning-based approach could be. I do not expect this baseline to be implemented in an update version as I believe it would be too much work for this iteration. But a discussion on what deep learning-based baseline that would be most suitable to compare to in future works would be beneficial.

2. The evaluation could be made more robust. Training on 200 samples and testing on 10 is reasonable, but generating a larger dataset would be useful to ensure that the performance estimates are reliable. It would also be interesting to see the variation in performance between different training runs to shed light on the stability of the proposed methodology.

3. The introduction could be made more friendly towards reader that are unfamiliar with the field of LU factorisation. The introduction starts with "The GMRES algorithm" without defining the acronym and without a reference. Similarly, LU factorisation is also presented without being defined. While both of these are well-known methods, it would increase the clarity of the writing if they were written out on their first appearance.

[1] J. Chen. “Graph Neural Preconditioners for Iterative Solutions of Sparse Linear Systems”. In: arXiv preprint arXiv:2406.00809 (2024)

**Final Rebuttal Confidence:**

4

**Final Rebuttal Justification:**

I think the authors have done a good job with the rebuttal. The addition of another data-driven approach is welcome, and it is also interesting to see the effect of different hyperparameters. Given my initial positive response, this has been reinforced and I keep my original recommendation.

**Justification:**

I think this is an interesting paper that is well-written with encouraging results. There are some limitations related to alternative baselines and the evaluation, but I do not consider these major limitations. Therefore, I recommend that the paper is accepted.

---

> ### Author Rebuttal · Authors · 2024-10-22
>
> We would like to thank the reviewer for their feedback and the positive evaluation of the paper. We improved the introduction chapter to make it more readable in the updated version.
>
> > Other deep learning-based baselines could improve the impression of the results.
>
> We added a comparison with previously developed data-driven incomplete Cholesky methods. The results can be found in Table 1. The implementation details for the method are in the appendix.
>
> > The evaluation could be made more robust.
>
> We added the training of several models in appendix D.2. for different hyper-parameters $\epsilon$ and added a discussion about the limitations of our experimental results in the future work section of the main paper.

---

### Official Review · Reviewer_h7hq · 2024-10-15
**Review of "Learning incomplete factorization preconditioners for GMRES"**

**Confidence:** 3

**Summary:**

This paper proposes to learn incomplete LU factorisations of sparse matrices through graph neural network data driven methods to be used a preconditioner for linear equations of GMRES. The paper highlights the flaws in traditional handcrafted methods that require significant time and strong heuristics to obtain a fast and accurate solutions for equation systems. This works proposes that the data driven approaches aim to learn tailored systems thus improving speed when adapting to different systems. The work is trained and evaluated on a synthetic dataset comprising problems arising from the discretization of the Poisson equation. Here results are presented for different losses and predconditioners. The results demonstrate improved performance both in terms of computational efficiency and task performance.

**Strengths:**

1. The paper is very well presented and written, while the flow allows for a pleasant read. The background presents enough information such that all readers can grasp key concepts of the paper, while the method explains the rationale clearly. Generally this work is very well presented.

2. Limitations are for the most part addressed, identifying issues regarding the distribution of problems and thus lack of generalisation to new problem domains. The issue of neural network training is also addressed.

3. The method itself provides a simple yet seemingly effective method to learn preconditioners. The method takes care to ensure desirable and essential properties are maintained such as invertibility, while doing so without overly elaborate mechanisms by employing a modified activation function.

4. The evaluation for the most part is effective at showing the computational performance of the approach while maintaining the task specific performance evaluated for.

5. The authors provide extensive details regarding the algorithmic implementation of their method that provides confidence in the reproducibility of the algorithm itself. All parameterisation of the method is also provided with extensive supplementary material.

**Weaknesses:**

Major Comments:

1. The clear weakness of this work lies in the experimental analysis. The use of one problem set, and synthetic data limits the understanding of method generalisation given there is an assumption that is not validated that the test data lies significantly outside the training distribution as to be a fair evaluation of performance. I understand the nature of this problem limits possible evaluation scenarios, however, further empirical analyses would significantly strengthen the work.
2. The proposed combined loss seemingly performs worse than individual losses but does achieve this result faster. However, the concern here is that the GMRES method with no preconditioner is also performing at a similar standard, yet much slower. Can the authors quantify more clearly how much benefit they obtain from their method given pre-training and performance gains not being drastic.
3. No comparisons made against existing works. Although, your related works clearly state there are other data driven methods to  generate preconditioners these are not empirically analysed. How do I know your method is better than these alternatives?
4. What is the distinct purpose of employing a graph neural network? Correct me if I’m mistaken, however, there seems to be no advantage or reason to employ a graph neural network over a standard ANN or CNN approach? While you employ a GNN to enforce the scarcity, the same effect can be achieved with other NN approaches and slightly modified constraints.

Minor Comments:

1. Your introduction could perhaps better explain the challenges and limitations of existing approaches to better define the problem statement.
2. The caption for Table 1 could be more informative, it is initially hard to interpret without careful reading of the main text. More effective caption could help the reader better interpret what metrics are being used to evaluate performance and what methods are being analysed. Also I assume time is in seconds? This needs clarification.
3. Given your method is primarily comparing against iterations and time, and as mentioned in your limitations, it would be beneficial to make more clear the compute time for pre-training and a cost (or at least approximate) of the time taken with pre-training in mind.
4. The background of graph neural networks could perhaps be omitted to the appendix given the audience of the conference, and the resulting space used to explain prior misconceptions and misunderstandings in further details.

Questions:
1. For the limits that you define as bounds of the training case to only explore he edges of the equation system. I understand that investigating the whole system is not tractable and your lemma provide the bounds, however, were any further analyses performed on cases outside of these bounds?
2. How does the impact of epsilon in the activation function affect performance of the method? How was this value found?

**Final Rebuttal Confidence:**

4

**Final Rebuttal Justification:**

The authors address my weaknesses and answer questions, given my already positive outlook I maintain my score.

**Justification:**

This work presents interesting approach that is simple in design yet effective in some settings. Although, more empirical analysis is needed to better validate the claims of the authors, the paper presents a novel method that is well grounded and justified in proof and literature. I therefore believe the contributions of this work to be valuable to the community and the level of novelty appropriate for the venue.

---

> ### Author Rebuttal · Authors · 2024-10-22
>
> We thank the reviewer for the feedback and their positive evaluation. We have updated the manuscript to improve the manuscript based on the suggested changes from the minor comments.
>
> > The clear weakness of this work lies in the experimental analysis.
>
> In the revised version we added additional experiments to strengthen the evaluation. However, we acknowledge that the numerical experiments in our presented work are limited. Therefore, we added a corresponding section to the future work section of our manuscript.
>
> > No comparisons made against existing works.
>
> We added a comparison with learned preconditioners for incomplete Cholesky methods to the main experiment section in our paper.
>
> > What is the distinct purpose of employing a graph neural network?
>
> Using graph neural networks is one of the fundamental building blocks of our method. GNNs have the following advantages compared to other architectures:
> - **Handle input of varying sizes**: not all matrices are of the same size so it is crucial to have a neural network that is trained once and then can be applied to any input size.
> - **Guaranteed sparse outputs**: While sparse CNNs have been developed they are not widely deployed. In particular, they do not guarantee the output to have a particular sparsity pattern. In contrast, using our proposed GNN model we can guarantee sparse outputs of the network
> - **Alignment**: GNNs and numerical linear algebra share a lot of common ground as previously shown [1]. In particular, the permutation equivariance of GNNs is connected with permutations of linear equation systems.
>
> **Questions**:
>
> 1. We are not entirly certain if we interpret the question correctly so please correct us if we are wrong. The main benefit of the shown lemmas is to provide a theoetrical understanding of existing loss functions that have been previously applied to train learned preconditioners. Based on these insights we are then able to come up with a novel loss functinos combining the previous approaches. We provide further analysis of the whole singular value distribution for the test problems (which the network does not see during training) in Figure 3 and in Appendix A.2. However, we did not manage to come up with a tractable loss function to take into account the whole spectrum.
>
> 2. We chose epsilon as a small constant to ensure invertability. In the updated manuscript we added additional experiments in Section D.2 where we compare the loss function for different values of the hyper-parameter epsilon.
>
> [1] N. S. Moore, E. C. Cyr, P. Ohm, C. M. Siefert, and R. S. Tuminaro. “Graph neural networks and applied linear algebra”. In: arXiv preprint arXiv:2310.14084 (2023)

---

### Meta-Review · Area_Chair_TCbh · 2024-11-02

**Recommendation:** Accept (Poster)
**Confidence:** 4

**Metareview:**

The reviewers are all convinced of accepting this paper.

**Suggested Changes To The Recommendation:**

2: I'm certain of the recommendation.  It should not be changed

---

### Decision · Program_Chairs · 2024-11-06

**Decision:**

Accept (Oral)

**Comment:**

We have decided to offer opportunities for oral presentations in the remaining available slots in the NLDL program. Thus, despite the AC's poster recommendation, we recommend an oral presentation in addition to the poster presentation given the AC's and reviewers' recommendations.